# Structures of ABCG2 under turnover conditions reveal a key step in the drug transport mechanism

Qin Yu [1], Dongchun Ni[2,3], Julia Kowal [1], Ioannis Manolaridis[1], Scott M. Jackson[1], Henning Stahlberg [2,4] & Kaspar P. Locher [1✉]

ABCG2 is a multidrug transporter that affects drug pharmacokinetics and contributes to multidrug resistance of cancer cells. In previously reported structures, the reaction cycle was halted by the absence of substrates or ATP, mutation of catalytic residues, or the presence of small-molecule inhibitors or inhibitory antibodies. Here we present cryo-EM structures of ABCG2 under turnover conditions containing either the endogenous substrate estrone-3-sulfate or the exogenous substrate topotecan. We find two distinct conformational states in which both the transport substrates and ATP are bound. Whereas the state turnover-1 features more widely separated NBDs and an accessible substrate cavity between the TMDs, turnover-2 features semi-closed NBDs and an almost fully occluded substrate cavity. Substrate size appears to control which turnover state is mainly populated. The conformational changes between turnover-1 and turnover-2 states reveal how ATP binding is linked to the closing of the cytoplasmic side of the TMDs. The transition from turnover-1 to turnover-2 is the likely bottleneck or rate-limiting step of the reaction cycle, where the discrimination of substrates and inhibitors occurs.

[1] Institute of Molecular Biology and Biophysics, Department of Biology, ETH Zürich, Zürich, Switzerland. [2] Center for Cellular Imaging and NanoAnalytics (C-CINA), Biozentrum, University of Basel, Basel, Switzerland. [3] Present address: Laboratory of Biological Electron Microscopy, Institute of Physics, SB, EPFL, Lausanne, Switzerland. [4] Present address: Laboratory of Biological Electron Microscopy, Institute of Physics, SB, EPFL, and Dep. Fund. Microbiol., Faculty of Biology and Medicine, University of Lausanne, Lausanne, Switzerland. ✉email: locher@mol.biol.ethz.ch

ABCG2 is an ABC transporter of broad substrate specificity expressed in various tissues and tissue barriers[1–4]. Among its endogenous substrates are steroids, including estrone-3-sulfate ($E_1S$), and the transporter has been reported to contribute to renal excretion of uric acid[5]. ABCG2 also functions as a multidrug transporter and exports a wide range of xenobiotics and pharmaceuticals, thus strongly impacting their pharmacokinetics[1–3]. Genetic polymorphisms of ABCG2 can lead to its dysfunction, which is associated with hyperuricemia and hypertension in humans[6]. Due to its broad substrate specificity, ABCG2 contributes to multidrug resistance in cancer[7,8]. For example, ovarian tumor and medulloblastoma cells over-expressing ABCG2 have shown increased resistance to chemotherapeutic agent including topotecan or mitoxantrone[9,10]. Considerable efforts have been directed at developing specific inhibitors of ABCG2 to counteract its activity in protecting tumor cells from anti-cancer drugs[11–17]. At present, the clinical applicability of such compounds is limited by their insufficient specificity, toxicity, or poor oral availability[18]. A detailed understanding of the mechanism of ABCG2 is therefore essential to advance the development of small-molecule modulators or inhibitors.

A series of initial structures have revealed the architecture of human ABCG2 and have provided insight into its binding of endogenous substrates as well as small-molecule inhibitors[19–21]. In the inward-open conformation, the pair of transmembrane domains (TMDs) form a slit-like cavity (cavity 1) that serves as a substrate-binding pocket. Two phenylalanine side chains, one from each ABCG2 monomer, clamp substrates between their phenyl rings[21–24]. This can rationalize that ABCG2 substrates are generally flat, polycyclic, and hydrophobic compounds[20]. Inhibitors can also bind in cavity 1, both competing with substrates for the binding pocket and interfering with the closing of the TMD dimer interface during the transport cycle[19–21]. Recent structural studies revealed that exogenous substrates (cytotoxic drugs used to treat various cancers) bind as single copies at the same general location as the endogenous substrate $E_1S$[22,23]. In addition to inward-open structures, the structure of a variant with reduced catalytic activity, ABCG2$_{E211Q}$, revealed an ATP-bound conformation with a closed nucleotide-binding domain (NBD) dimer and a collapsed translocation pathway. This was taken as evidence supporting an ATP-driven, peristaltic extrusion mechanism[21].

The common denominator of all published ABCG2 structures is that they represented trapped states determined under conditions where the transporter was prevented from cycling through its catalytically relevant conformations. This was accomplished either by the removal of ATP or transport substrates or by the addition of small-molecule inhibitors or externally binding inhibitory Fab fragments such as 5D3-Fab[19,25]. The strategy of locking intermediate states or reducing conformational flexibility has been at the heart of most high-resolution structural studies of multidrug ABC transporters since the publication of the Sav1866 structure[26]. Mutation of the catalytically essential Walker-B glutamate has often been used to trap ATP-bound states, where the nucleotide is sandwiched between the NBDs[21,27]. Occasionally, disulfide cross-linking was employed to trap intermediate states[28,29]. Such approaches were essential when using X-ray crystallography as a structural technique, since excessive conformational dynamics interfere with the generation of well-ordered 3D crystals. On the downside, these conditions differed from the physiological environment, where substrates, ATP, ADP, and $Mg^{2+}$ are all present. As a result, the catalytic cycle of ABCG2 and other multidrug transporters is insufficiently understood. While conformational rigidity is also beneficial for cryo-EM approaches, the single-particle nature of cryo-EM studies provides an opportunity to investigate multidrug ABC transporters under turnover conditions, where distinct conformations are allowed to co-exist[30]. This approach is essential for understanding the transport mechanism because it reduces the chance of visualizing physiologically irrelevant conformational states or intermediates.

In the present study, we reconstitute human ABCG2 in lipid nanodiscs, which provide a near-native environment containing phospholipids and cholesterol. We apply turnover conditions using two distinct substrates and find a single major turnover state for the endogenous steroid $E_1S$ and two states for the larger, exogenous drug topotecan. Our study reveals key conformational changes that are essential for substrate recognition and translocation and allow ABCG2 to distinguish substrates from inhibitors.

## Results

**Structures of ABCG2 under turnover conditions.** We determined the ATPase activity of ABCG2 reconstituted in nanodiscs (Supplementary Fig. 1a) in the presence of $E_1S$ or topotecan (Fig. 1a). To ascertain that the two substrates bound to nanodisc-reconstituted ABCG2, we measured the modulation of the ATPase rate compared to the absence of substrates. Compared to liposome-reconstituted ABCG2, the basal ATPase rate in nanodiscs is markedly elevated, as was observed previously[19–21]. This rate was further increased in the presence of $E_1S$, but decreased in the presence of topotecan (Fig. 1a). While the absolute values are higher than in liposomes, this observation, combined with the structural data, suggests that the two substrates indeed bound to nanodisc-reconstituted ABCG2. To mimic turnover conditions, we incubated ABCG2 with 5 mM ATP, 0.5 mM ADP, 5 mM $MgCl_2$, and either 200 μM $E_1S$ or 100 μM topotecan, and applied the mixtures to EM grids. We collected single-particle cryo-EM data of these two samples, which allowed us to determine three distinct high-resolution structures, one from the $E_1S$-bound ABCG2 and two from topotecan-bound ABCG2. Both the NBDs and TMDs were well resolved in all three structures (Supplementary Fig. 1b). Despite the fact that ABCG2 is a homodimer, we refined all maps in C1 symmetry and did not observe significant structural differences between the two monomers.

The turnover sample containing $E_1S$ revealed a single, well-defined conformation resolved at 3.4 Å resolution (Supplementary Fig. 2). In contrast, the turnover sample containing topotecan revealed two well-ordered structures, which we termed turnover-1 and turnover-2 states, resolved at 3.1 Å and 3.4 Å, respectively (Supplementary Fig. 3). The conformation of the $E_1S$-bound structure was very similar to turnover-2 from the topotecan-containing sample (Supplementary Table 1). We, therefore, refer to the $E_1S$-bound structure as turnover-2 as well. The TMDs in the turnover states adopted inward-facing conformations but differed in their degree of opening towards the cytoplasmic side of the membrane. Turnover-1 is a more open conformation, whereas turnover-2 is a more closed conformation. All three structures revealed densities for transport substrates and two bound nucleotides, which were interpreted as bound ATP molecules (Fig. 1b). This choice was made because the relevant density in turnover-2 suggests bound ATP, while modeling ADP would leave unexplained, additional density (Supplementary Fig. 4). For the turnover-1 state, the density of bound nucleotides is weaker because there are fewer contacts with the protein. While it would in principle be possible to interpret the density as a mixture of ADP and ATP, we built ATP because it did not violate the observed density features and because the concentration of ATP in the solution is ten times higher compared to that of ADP.

While we assume that a closed conformation similar to that observed in the structure of the ATP-bound ABCG2$_{E211Q}$ variant

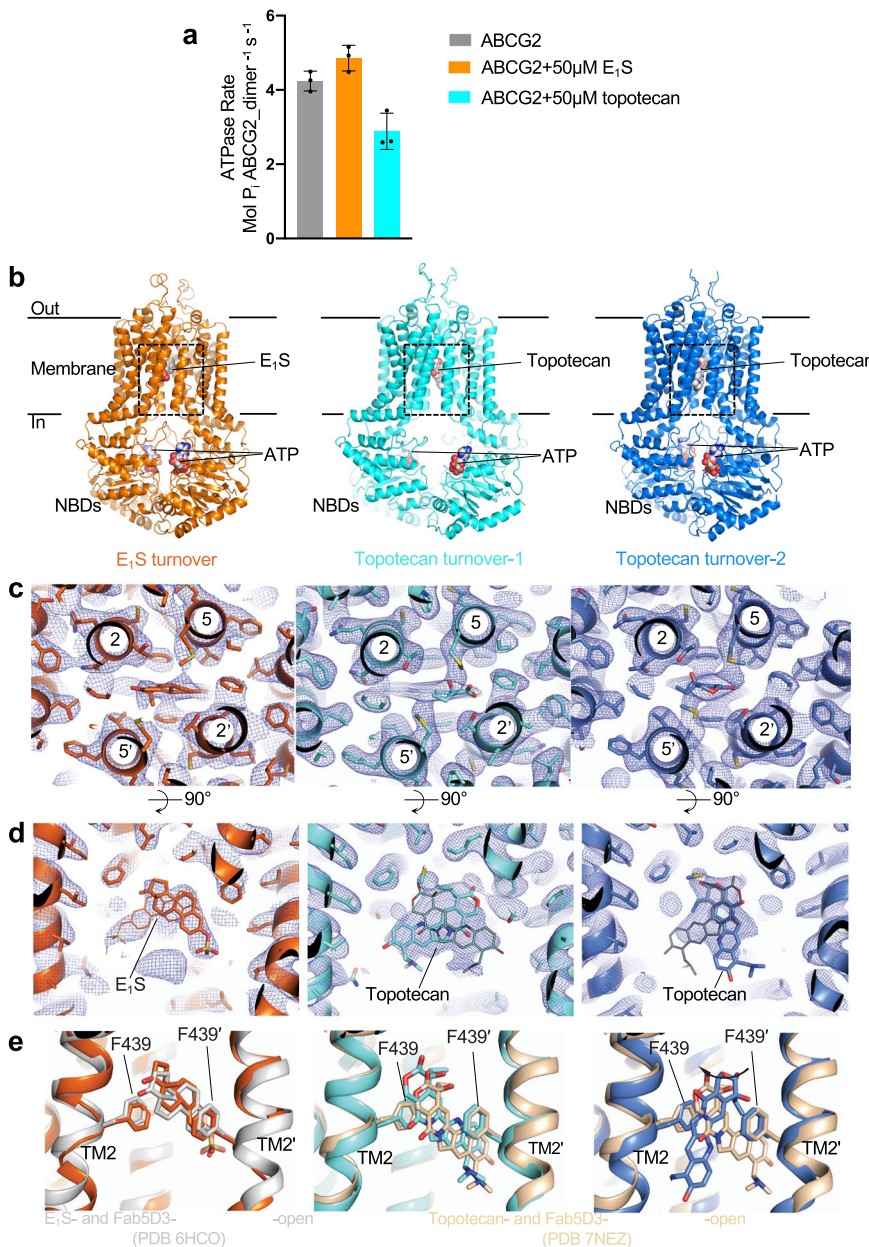

**Fig. 1 Structures of ABCG2 under turnover conditions. a** ATPase activity of nanodiscs-reconstituted wild-type ABCG2 in the presence and absence of 50 µM $E_1S$ or 50 µM topotecan. The bars show means rate. The experiment was performed three times independently with the same batch of nanodiscs ($n = 3$); error bars depict standard deviations (s.d.) Source data are provided as a Source Data file. **b** Ribbon diagrams of turnover structures. $E_1S$ turnover is shown in orange, topotecan turnover-1 is shown in light cyan and topotecan turnover-2 is shown in blue. The coloring of the three structures is maintained throughout the figures and panels. Bound ATP, $E_1S$, and topotecan are shown in sphere representation and labeled. **c** Close-up view of substrate-binding pockets between the TMDs viewed from the extracellular side. TM helices are shown as ribbons and labeled, residues and bound substrates are shown as sticks. Substrates are at center of the panels. Non-symmetrized EM density maps are shown as blue mesh. **d** Similar to **c**, but viewed from within the membrane. Bound substrates are labeled. Two possible orientations of bound substrates are shown for each density as thin or thick sticks, respectively. **e** Superposition of turnover states with structures of substrate-bound inward-open ABCG2 bound to 5D3-Fab fragments using the side chains of F439s as anchors. Left, $E_1S$-bound ABCG2 structure (PDB 6HCO) is shown in gray cartoon. Middle and right, topotecan-bound ABCG2 structure (PDB 7NEZ) is shown in yellow cartoon. The substrates and side chains of F439 residues are shown as sticks.

is present under turnover condition[21], we did not observe a defined class of such particles in our turnover samples. This suggests that this closed conformation is not a low-energy state of the wild-type protein under turnover conditions. We also did not observe a collapsed apo-state similar to that reported recently for apo-ABCG2[22]. In the topotecan-bound ABCG2 sample, the dominant class (88% of the ordered particles) belonged to the turnover-1 state, whereas 12% particles belonged to the turnover-2 state (Supplementary Fig. 3). This suggests that turnover-1 is the lowest-energy state of ABCG2 in the presence of topotecan, whereas turnover-2 is the lowest-energy state in the presence of $E_1S$. In all turnover structures, the leucine gate (formerly called leucine plug), which separates cavity 1 from cavity 2, is closed (Supplementary Fig. 5a, b).

**E₁S and topotecan binding site.** We initially expected that under turnover conditions, transport substrates would either be too mobile to be observed or not present at all. To our surprise, we found strong densities for bound substrates lodged in cavity 1 on the twofold molecular symmetry axis of ABCG2 (Fig. 1c, d). Substrate recognition and binding are similar as in the previously reported, inward-open structures[21–23]. The polycyclic cores of E₁S and topotecan are sandwiched between the F439 side chains of the two ABCG2 protomers (Fig. 1c, d). Only one substrate molecule can be fitted inside the density, as fitting two molecules would introduce serious steric clashes (Fig. 1d). The density suggests a certain degree of flexibility of bound substrates, in particular for topotecan (Fig. 1d). In the structure containing E₁S (turnover-2), the substrates could be fit in two orientations related approximately by a 180° rotation. Due to the C1 processing of the data, one orientation is more prominent. When compared to the previously reported structure of E₁S- and 5D3-Fab-bound ABCG2 in an inward-open conformation, E₁S appeared slightly shifted (~1 Å) towards the leucine gate and thus towards the external side of the transporter (Fig. 1e). In the topotecan-containing turnover structures, the shape of the density feature covering the drug is different in turnover-1 versus turnover-2. In turnover-1, topotecan can be fitted in two orientations related approximately by a 180° rotation (Fig. 1d middle), similar to what was observed in an earlier structure of topotecan- and 5D3-Fab-bound, inward-open ABCG2[23], and with no significant shift of bound topotecan in the direction of the leucine gate. In turnover-2, the topotecan density was narrower in the horizontal dimension and longer in the vertical dimension, probably reflecting a rotation of the drug compared to the turnover-1 state (Fig. 1d right). The changes in EM density could be visualized by 3D variability analysis of the final particles using the program cryoSPARC (Suppl. movies 1 and 2)[31,32].

**Coupled NBD and TMD motions.** ABC transporters use the energy of ATP binding and hydrolysis to translocate substrates across the membrane via alternating access mechanism[33–35]. In the course of the ATPase cycle, the NBDs cycle between closed and open conformations to hydrolyze ATP or to release ADP and phosphate. This drives conformational changes in the TMDs. The separation of the NBDs in our turnover-1 and turnover-2 structures is smaller than that in the fully inward-open, topotecan-bound ABCG2 structure[23], but larger than that in the ATP-bound structure of ABCG2_{E211Q} (Fig. 2a)[21]. A comparison of these four structures shows that as the NBDs approach each other, so do the cytoplasmic sides of the TMDs (Fig. 2a). The TMDs rotate relative to the NBDs (Supplementary Fig. 5c). Compared to the inward-open conformation, access to cavity 1 from the inner leaflet of the membrane is reduced in turnover-1, as the gap between transmembrane helices TM1 and TM5' of the opposing ABCG2 monomer is narrower (Fig. 2b, c). In turnover-2, the gap is even smaller, causing the entrance to cavity 1 from within the membrane to be completely sealed and even the opening to the cytosol to be reduced. As a result, a larger substrate such as topotecan could not enter into, or exit from, cavity 1 without spreading of the cytoplasmic sides of the TMDs (Fig. 2c). The degree of NBD closing and TMD gap narrowing can be measured by comparing the distances of the coupling helices (CpH), which are located in the first intracellular loop of the TMDs and transmit conformational changes at the NBD-TMD interface. In turnover-1, the coupling helices are moved towards each other by ~1 Å compared to the fully inward-open, topotecan- and 5D3-Fab-bound structure (Fig. 2d). In turnover-2, the distance between the coupling helices is reduced by another 4.0 Å. The distances between the TMDs have shifted accordingly

(Supplementary Fig. 6). This reveals a strict coupling between NBD and TMD closing.

In addition to the motion of the NBDs as full domains, we observed a rotation of the RecA-like and the helical sub-domains relative to each other, which caused the helical sub-domain to further approach the opposite NBD. Superimposing the RecA-like domains and measuring the relative angles of the α-helical domains revealed rotations of 2°, 19°, and 24° between the fully inward-open conformation and turnover-1, turnover-2, and the ATP-bound closed state (Supplementary Fig. 7).

A comparison of the topotecan-bound turnover-1 and turnover-2 structures revealed a key NBD segment contributing to functionally relevant NBD–NBD and NDB–TMD domain contacts. As ABCG2 transitions from turnover-1 to turnover-2, the segment between Q181 and V186 undergoes a conformational change and facilitates key inter-domain contacts (Fig. 3a, b). V186 is the first residue of the VSGGE motif (equivalent to LSGGQ more commonly found in ABC transporters)[36]. This motif is among the most conserved in ABC transporters and pins bound nucleotides against the Walker-A and Walker-B motifs of the opposing NBD. We now found that in turnover-2, Q181 and F182 facilitated contacts between TM1 of one ABCG2 monomer and TM5' of the other (Fig. 3a, b), thus helping stabilize the TMD domain closure. In addition, the arginine residue R184 provides a direct contact, in the form of a cation-π interaction, with the adenine moiety of the ATP molecule bound to the opposite NBD (Fig. 3d). Neither of these contacts is formed in the turnover-1 structure (Fig. 3c).

We analyzed the role of R184 by mutating it to an alanine (Supplementary Fig. 8a, b). We reconstituted the purified ABCG2_{R184A} variant in proteoliposomes and performed ATPase and transport assays (Fig. 3e, f). We found that in line with its location in the structure, both the ATPase and transport activities of ABCG2_{R184A} were greatly reduced. As was observed for WT ABCG2, the ATPase rate of ABCG2_{R184A} was stimulated by E₁S and topotecan (Fig. 3e). Furthermore, the E₁S transport rate was similarly reduced for WT ABCG2 and ABCG2_{R184A} in the presence of topotecan, suggesting that the mutation of R184 to alanine neither interfered with substrate binding nor abolished the NBD-TMD coupling (Supplementary Fig 9a). Given that the EC_{50} values of E₁S- and topotecan-induced stimulation of the ATPase activity of ABCG2 were similar (15.7 μM and 11.7 μM, respectively, Supplementary Table 2)[21,23], the reduction of the initial E₁S transport rate by ~50% in the presence of topotecan is in line with the structural data in that topotecan and E₁S bind and compete for the same binding pocket both in WT ABCG2 and in the ABCG2_{R184A} variant (Supplementary Fig. 9b).

**The role of R482.** The single-nucleotide polymorphisms R482G and R482T, originally cloned from drug-resistant cancer cell lines[37,38], have been reported to alter the substrate specificity of ABCG2[39]. Cells expressing ABCG2_{R482G} were found to efficiently extrude rhodamine-123 or doxorubicin but were less resistant to topotecan, an anticancer drug that inhibits topoisomerase[7,38,40,41]. It was, therefore, suggested that R482 influenced substrate transport and ATP hydrolysis, but not substrate binding[42,43]. R482 is located in TM3 and does not directly contact bound drugs[21–23]. Rather, it contacts TM2 which contains the key residue F439 that interacts with substrates in cavity 1[19–21]. Intriguingly, we found that the side chain of R482 adopts distinct conformations in the turnover-1 and turnover-2 structures (Fig. 4). In turnover-1, the side chain points towards the cytoplasmic side of the membrane, where the guanidinium group forms a hydrogen bond with the side-chain of S443 of TM2 (Fig. 4b and c, left). In contrast, the R482 side chain is rotated towards the external side of the membrane in turnover-2, where

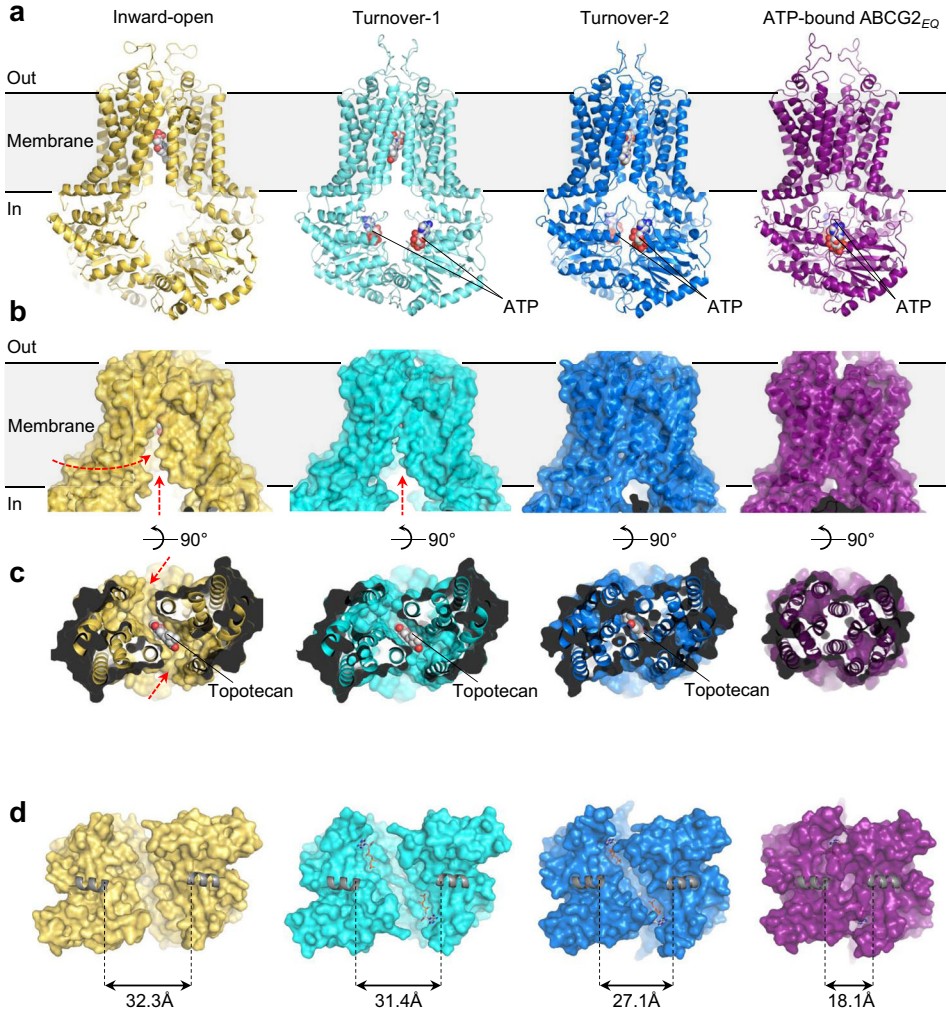

**Fig. 2 Coupled NBD and TMD closing.** Structures of four states are shown: "Inward-open" depicts topotecan- and 5D3-Fab- bound ABCG2 (PDB ID 7NEZ, colored in yellow), the turnover states are from this study, "ATP-bound ABCG2$_{E211Q}$" depicts the closed conformation (PDB ID: 6HBU, colored in purple). **a** Ribbon diagrams with substrates and ATP shown as spheres. **b** Surface representations of ABCG2, emphasizing the distinct TMD conformations. Red arrows depict openings to cavity 1 from within the lipid bilayer or from the cytosol. **c** View of substrate-binding site in cavity 1 of ABCG2. The TMDs are shown in ribbon and surface representation and viewed from the cytoplasmic side of the membrane. Substrates are shown as spheres and labeled. Red arrows depict openings to cavity 1 from within the lipid bilayer. **d** Surface representations of the NBDs viewed from the membrane. Bound ATP is shown as sticks. The coupling helices (CpHs) are shown as gray cartoons. Distances between CpHs are indicated.

the guanidinium group contacts side chains and main-chain atoms from TM2 and TM4 in the direct vicinity of F439 (Fig. 4b, c, right). This suggests a structural role for R482 in facilitating the conformational changes required to transition from turnover-1 to the turnover-2 state (visualized in Suppl. movie 2). The impact of mutations of R482 on drug recognition is therefore likely of an allosteric nature.

**Steric clashes of inhibitors in turnover-2 state.** Our structures provide additional insight into the mechanism of small-molecule inhibitors of ABCG2. Previous structures revealed that the inhibitors MZ29 (a derivative of Ko143) and MB136 (a derivative of tariquidar) bound to cavity 1 of inward-open ABCG2, which was bound to 5D3-Fab[20]. We find that while the size of cavity 1 in our turnover-1 structure is slightly smaller than that of the fully inward-open conformation, only minor steric clashes would exist with the tert-butyl group of MZ29. In contrast, the strongly reduced size of cavity 1 in the turnover-2 conformation would lead to major steric clashes with bound inhibitor molecules. Figure 5 reveals that whereas there is sufficient space in turnover-

2 for bound substrates (E$_1$S or topotecan), the inhibitors MZ29 and MB136 could not fit. The concerted motions of TM1, TM2, and TM5 reduce the volume of cavity 1 by approximately one-third, from ~1300 Å$^3$ in turnover-1 to ~830 Å$^3$ in turnover-2[44], which is incompatible with the binding of the inhibitor molecules. We conclude that with small-molecule inhibitors bound, ABCG2 may be able to adopt the conformation of turnover-1, but could not adopt the conformation of the turnover-2 state. Unlike in B-subfamily ABC transporters, where longer intracellular loops allow additional flexibility, binding of inhibitors to ABCG2 prevents the TMDs to close at the cytoplasmic side and consequently the NBDs from forming a closed dimer arrangement, which is required for ATP hydrolysis. This can explain why ABCG2 has a strongly reduced ATPase rate in the presence of inhibitors[20], whereas ABCB1 does not[20,29,45,46].

## Discussion

In contrast to earlier structural studies of ABCG2, we could not predict whether the transporter would adopt a single conformation or multiple distinct conformations under turnover

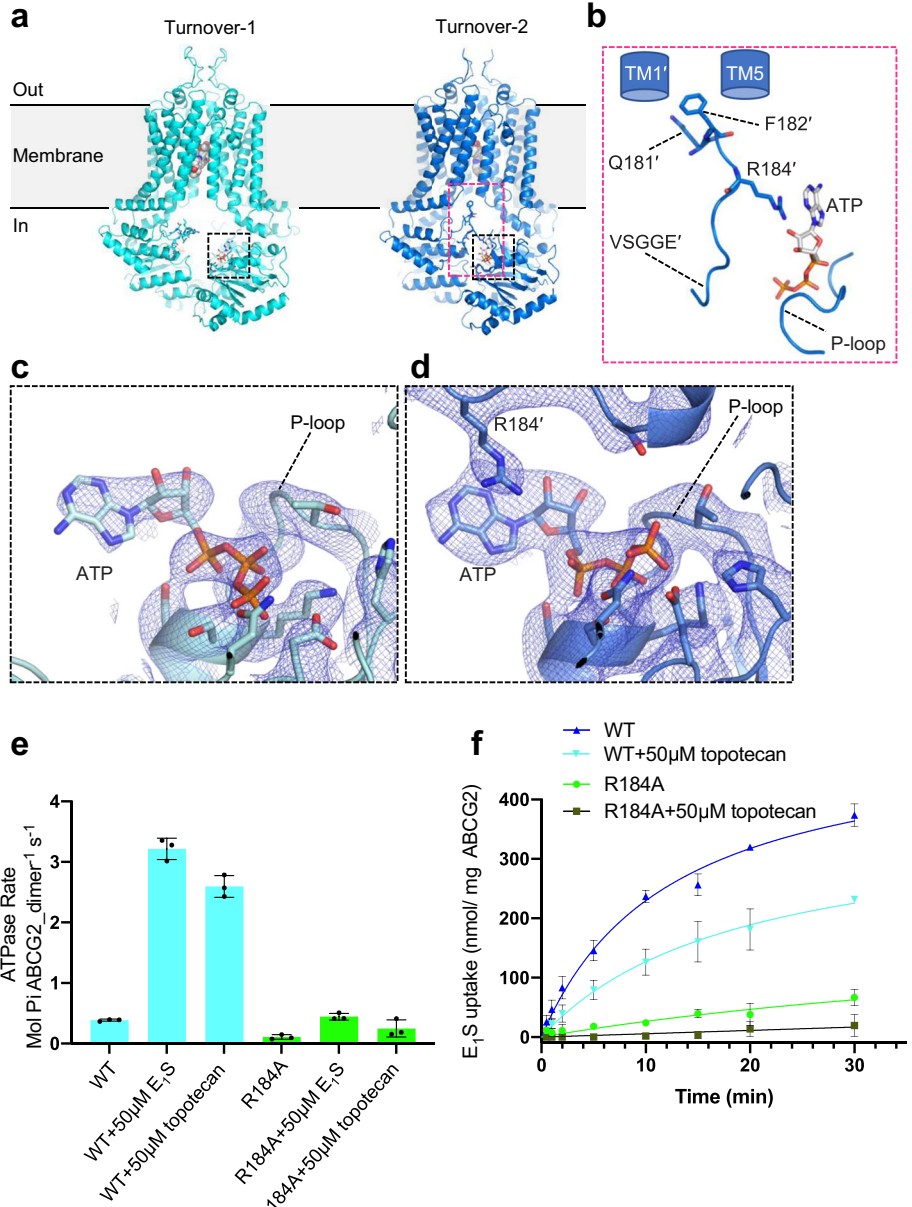

**Fig. 3 Domain interfaces and the role of R184. a** Ribbon diagrams of topotecan-bound turnover states. ATP is shown as sticks, topotecan is shown as spheres. Dashed black and red boxes show domain interfaces with close-up views in **b**–**d**. **b** Close-up view of domain interface in turnover-2 conformation. TM1' and TM5 are schematically shown as blue cylinders. P-loop, X-loop, and signature motif (VSGGE) are shown in cartoon mode. ATP and key side chains involved in surface interactions are shown as sticks. **c** and **d** Close-up views of ATP-binding sites of turnover-1 and turnover-2 states, as indicated in **a**. ABCG2 is shown as a ribbon diagram, key residues and bound ATP are shown as sticks and labeled. Non-symmetrized EM density maps are shown as blue mesh. **e** ATPase activity of liposome-reconstituted wild-type and mutant ABCG2 (R184A) in the presence and absence of 50 µM $E_1S$ or 50 µM topotecan. Data are presented as mean rates ± s.d. The experiment was performed 3 times independently with the same batch of liposomes ($n = 3$). Error bars depict s.d. **f** ABCG2-catalyzed $E_1S$ transport into proteoliposomes by wild-type ABCG2 or R184A variant in the presence or absence of 50 µM topotecan. Data are presented as mean values ± s.d. The experiment was performed 3 times independently with the same batch of liposomes ($n = 3$). Error bars depict s.d. Source data are provided as a Source Data file.

conditions. We also could not predict which state would correspond to the lowest energy state in the transport cycle. Our finding that two distinct inward-open conformations containing both a bound transport substrate and two ATP molecules corresponded to the lowest energy states was nevertheless unexpected. This arrangement was often considered to be of higher energy or even transient in ABC transporters, and thus expected to quickly convert to intermediates where either the transport substrate was absent, or ATP had been hydrolyzed[47,48]. Indeed, an analysis of bovine ABCC1, another multidrug resistance

protein, captured a post-hydrolysis state under turnover conditions featuring a closed NBD dimer and a (slightly) outward-open substrate translocation pathway. This resembled a previously reported structure of a catalytically impaired ABCC1 variant containing a Walker-B glutamate to glutamine mutation[47]. In contrast to bovine ABCC1, a bacterial ABC transporter of peptides, TmrAB, was found to adopt multiple inward-open conformations under turnover conditions. These varied in how widely the NBDs were separated, the degree of inward opening of the translocation pathway, and whether bound nucleotides or

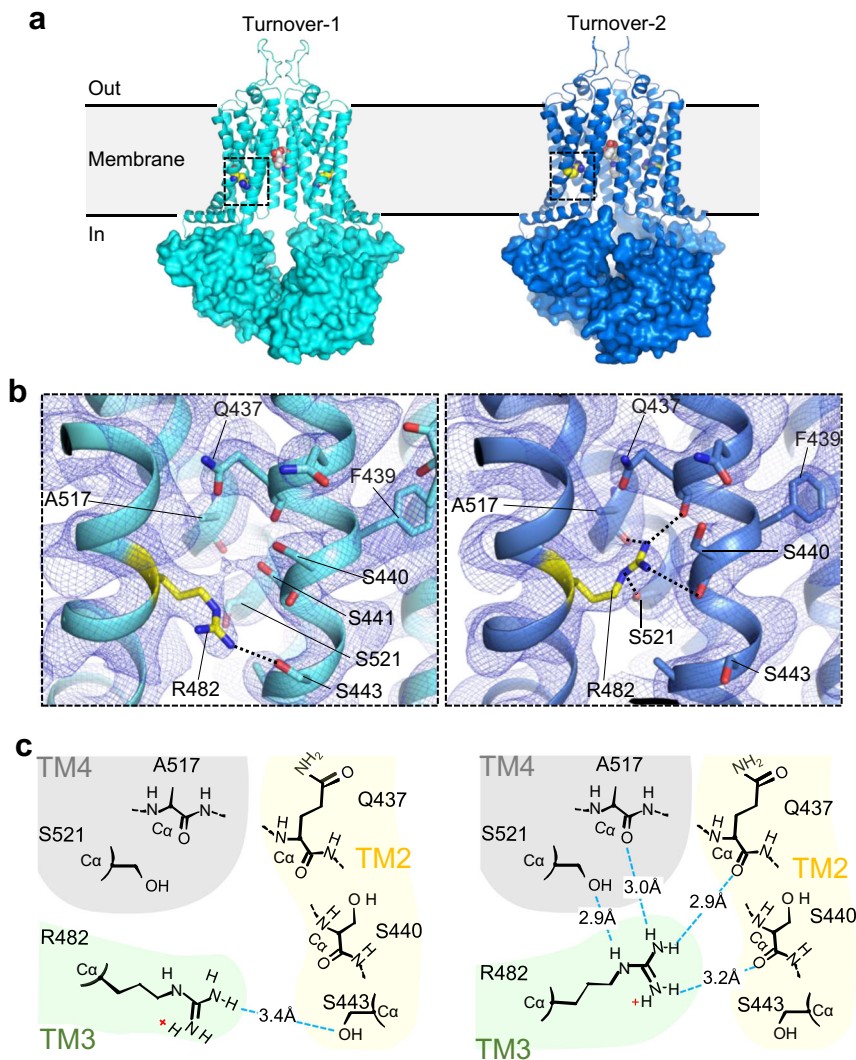

**Fig. 4 Conformational change of R482 in turnover states. a** Ribbon diagrams of topotecan-bound turnover states, with NBDs shown as surfaces. Bound topotecan and the side chain of R482 are shown as spheres. Dashed black boxes show regions around R482. **b** Closeup views of TMD regions indicated in **a**. ABCG2 is shown as a ribbon diagram in turnover-1 (left) and turnover-2 (right) states. R482 and surrounding residues are shown as sticks and labeled. Non-symmetrized EM density is shown as blue mesh. Hydrogen bonds are indicated as black dashed lines. **c** Schematic diagram of hydrogen bonds formed by R482 in turnover-1 (left) and turnover-2 (right). H-bonds are shown as blue dashed lines. Yellow shading highlights residues of TM2, green shading highlights residues of TM3, gray shading highlights residues of TM4.

substrates could be identified[48]. Our findings with human ABCG2 are distinct from both of those studies, which most likely reflects the mechanistic diversity within the ABC transporter superfamily[34].

By combining previously reported structures of trapped states and our turnover structures presented here, we can now propose a structure-based hypothesis of the complete transport cycle of ABCG2 (Fig. 6). Only conformational states that are required for transport are included in this scheme. Our turnover structures have a key role in this mechanistic scheme. Given that $E_1S$ did not promote the turnover-1 state, we concluded that ABCG2 can directly proceed to turnover-2 when smaller or endogenous substrates are bound. We anticipate a similar situation with urate, an even smaller molecule than $E_1S$. For these substrates, turnover-2 appears to immediately precede the rate-limiting step of the transport cycle. In contrast, turnover-1 was the dominant class in the topotecan turnover sample. This suggests that the transition from turnover-1 to turnover-2 may be the step where exogenous compounds are "tested" for their suitability as transport

substrates. Because topotecan is larger than $E_1S$, the transition from turnover-1 to turnover-2 is slower. We conclude that the degree of TMD and NBD closing in the lowest energy intermediate correlates with the shape and size of ABCG2 substrates. More generally, compounds that bind in cavity 1 and allow ABCG2 to adopt the turnover-2 conformation may be transported, even if their transport rates are slower than that of $E_1S$. In contrast, in the presence of strongly bound inhibitors such as Ko143 and its derivatives[13,20], ABCG2 cannot advance to the turnover-2 conformation.

Our results further assign a key role to the segment 181-184 of ABCG2. It was previously reported that the polypeptide segment leading up to the LSGGQ motif appeared to have a role in stabilizing TMD-TMD interfaces in B-subfamily ABC transporters, and the term X-loop was introduced to account for this role[26]. Rather than an aromatic residue to stabilize the adenine moiety of bound nucleotides, as observed in the A-loops of most ABC transporters[27,49–51], ABCG2 relies on R184 from the opposite protomer to assume this function. R184, therefore, appears to

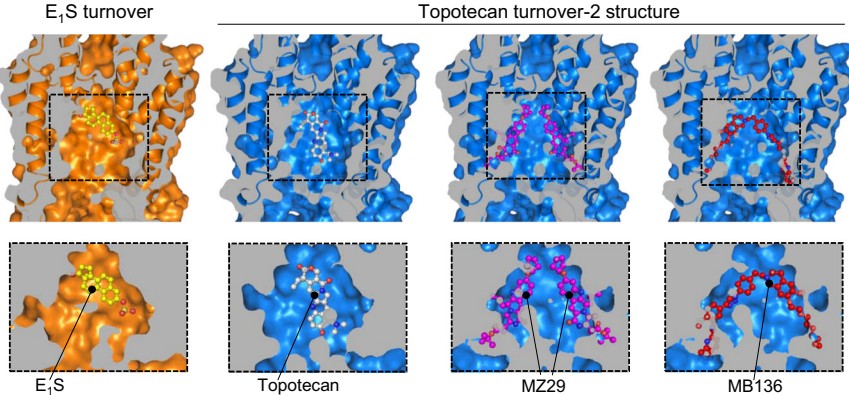

**Fig. 5 Cavity 1 in turnover-2 state accommodates substrates but not inhibitors.** Vertically sliced views through $E_1S$-bound (left) and topotecan-bound (right) turnover-2 structures. The top row of panels shows the TMDs both as ribbons and in surface representation. The bottom row shows close-up views of cavity 1, with ABCG2 in surface representation. Bound $E_1S$ and topotecan are shown in mixed stick/sphere representation as built in the structures. Inhibitor structures were extracted from inward-open ABCG2 structures containing these inhibitors (PDB 6ETI and 6FEQ, respectively). To place inhibitors into cavity 1 of the turnover-2 state, we aligned the phenyl moieties of the two F439 residues in PDB 6ETI or in 6FEQ with those of our turnover structures. In this way the translocation pathway of PDB 6ETI, 6FEQ, and turnover-2 are well-aligned and inhibitors can be fit. The inhibitors placed into cavity 1 of the topotecan turnover-2 emphasize the resulting steric clashes.

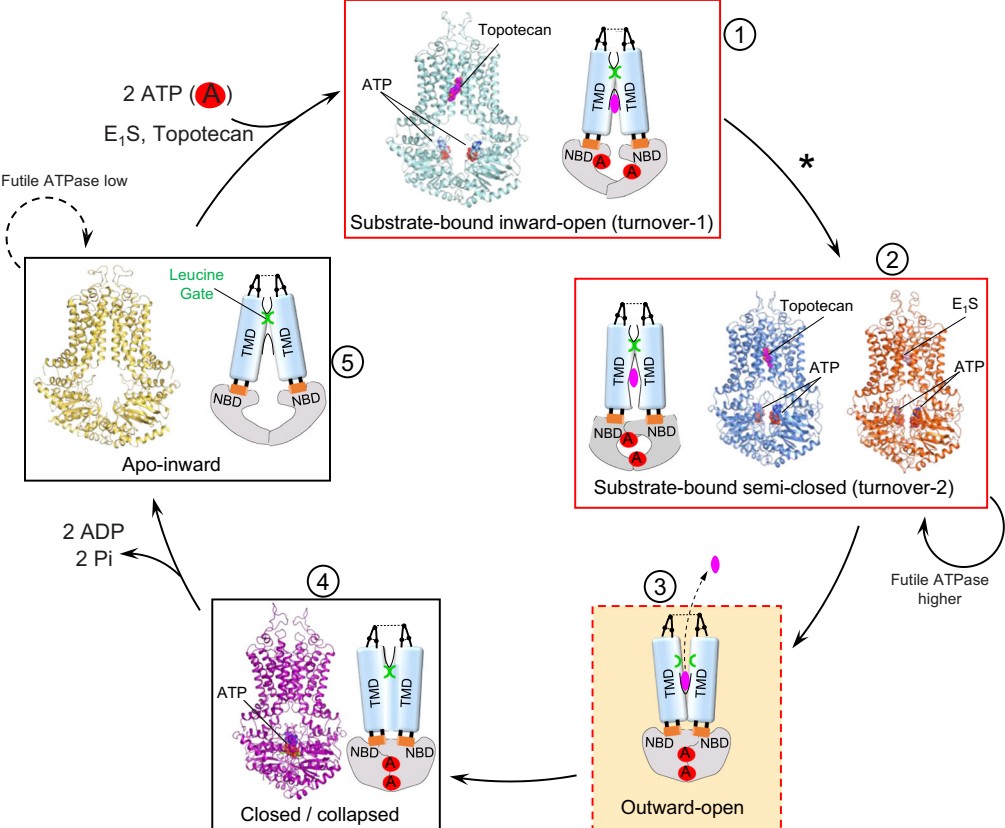

**Fig. 6 Transport mechanism of ABCG2.** Schematic of structure-based, proposed transport cycle of ABCG2. Key states are shown as boxes and numbered. Where available, structures are shown as ribbon diagrams with bound nucleotides and substrates shown as spheres and labeled. Cartoon schematics of the relevant states are drawn to emphasize the conformations of the TMDs and NBDs, identify cavities, indicate the presence of bound drugs, and as well as the state of the leucine gate. Previously determined structures correspond to PDB IDs 5NJ3 (apo-inward) and 6HBU (closed conformation). No structure is available of the proposed outward-open conformation (state 3).

have a dual role in the turnover-2 state: It pioneers the contact between the two NBDs while simultaneously strengthening the binding of ATP through a cation-π interaction with the adenine moiety. The initial NBD-NBD contact is probably important because the signature motif is not yet close enough to interact with the ribose and triphosphate moieties of the ATP molecule.

Our mechanistic scheme indicates two distinct levels of ATPase activity, a basal and a stimulated level (Fig. 6). In the absence of substrates, ABCG2 displays a basal ATPase rate that generates 0.65 molecules of phosphate per ABCG2 dimer per second. In the presence of $E_1S$, the ATPase rate increases to 2.4. We compared this value with the initial $E_1S$ transport rate, which is 0.1

molecules $E_1S$ per ABCG2 dimer per second. If ABCG2 hydrolyzes two molecules of ATP in one productive transport cycle, the net futile ATPase rate in the presence of $E_1S$ is 2.2 molecules phosphate per ABCG2 dimer per second, which is 3.4 times higher than in the absence of $E_1S$. We conclude that substrates not only increase the ATPase rate of ABCG2 due to productive transport, but also increase the basal/futile ATPase rate. This suggests that ABCG2 may require multiple attempts (each consuming ATP) before substrate extrusion is successfully accomplished.

In both turnover states described here, the leucine gates are closed. This is notable in the case of turnover-2, where not only the NBDs are semi-closed, but the cytoplasmic ends of the TMDs are also partially closed. This suggests that the opening of the leucine gate may only occur once the cytoplasmic ends of the TM helices are further pushed together and peristaltic pressure is exerted on bound substrate. We speculate that at this point, ABCG2 must adopt an outward-facing conformation, in which the leucine gates are opened and substrate can escape via cavity 2 into the extracellular environment. Such a conformation may be transient in nature given that we have not observed it in our turnover samples. Further studies are required to capture ABCG2 or an appropriate ABCG2 variant in an outward-open conformation.

In conclusion, our results provide key insight into the transport cycle of ABCG2 and help understand how small-molecular compounds act as substrates or inhibitors. Our study may also have predictive value for understanding the mechanisms of other ABC transporters, in particular of the G-subfamily. Given their diversity with respect to mechanistic details, it is quite possible that ABC transporters belonging to other subfamilies will reveal distinct conformations or intermediate states most strongly populated under turnover conditions.

## Methods

**Expression and purification of ABCG2**. Human wild type ABCG2 (Uniprot: Q9UNQ0) or ABCG2 $_{R184A}$ containing an N-terminal Flag-tag was expressed in HEK293-EBNA (Thermo Fisher Scientific) cells by transient transfection[19]. Cells were incubated at 37°C for 48–60 h before harvesting. Harvested Cell were lysed using a Dounce homogenizer and solubilized with 1% DDM, 0.1% CHS (cholesteryl hemisuccinate) (w/v) (Anatrace), 40 mM HEPES buffer pH 7.5, 150 mM NaCl, 10% (v/v) glycerol, 1 mM PMSF (phenylmethylsulfonyl fluoride), 2 μg ml$^{-1}$ DNaseI (Roche), and protease inhibitor cocktail (Sigma). Lysed cells were centrifuged at 100,000 g and the supernatant was incubated with anti-Flag M2 affinity agarose gel (Sigma). ABCG2 was eluted with Flag peptide (Sigma) and applied to a Superdex 200 increase 10/300 column (GE Healthcare) in 40 mM HEPES, pH 7.5, 150 mM NaCl, 0.026% DDM and 0.0026% CHS (w/v). Peak fractions were collected for further use.

**ABCG2-nanodisc preparation**. Membrane scaffold protein (MSP) 1D1 was expressed in E. coli and purified as described[52]. ABCG2 reconstitution in nanodiscs was performed as following methods[19]. In brief, brain polar lipid (BPL, Avanti Polar Lipids) and CHS (cholesteryl hemisuccinate) were mixed at a 4:1 (w/w) ratio. Lipids were solubilized with a 3x molar excess of sodium cholate using a sonic bath. Lipids were then mixed with MSP1D1 and detergent-purified ABCG2 at a molar ratio of 100:5:0.2 (lipid:MSP:ABCG2). Detergent was removed by addition of Bio-Beads SM-2 (Biorad) and incubated at 4 °C overnight. Biobeads were removed and the sample was centrifuged at 100,000 g for 30 min. The supernatant was loaded on a Superdex 200 increase column and fractions containing nanodisc-reconstituted ABCG2 were collected for Cryo-EM grid preparation or for functional assays.

**ABCG2-liposome preparation**. ABCG2-containing proteoliposomes were prepared as described[19,53]. In brief, BPL was mixed with cholesterol (Chol) at a 4:1 (w/w) ratio. Liposomes were resuspended in transport buffer (25 mM HEPES pH 7.5, 150 mM NaCl) and extruded using a 400 nm polycarbonate filter (Avanti Polar Lipids). To reconstitute ABCG2, liposomes were destabilized with 0.17% (v/v) Triton X100. Detergent-purified ABCG2 was mixed with liposomes at a 100:1 (w/w) lipid: protein ratio. Detergent was then removed using Bio-Beads, added in multiple batches. Proteoliposomes were collected using centrifugation at 100,000 g. The proteoliposome pellet was resuspended in transport buffer at a final lipid concentration of 10 mg ml$^{-1}$.

**ATPase and transport assays**. ATPase assays were performed at 37 °C in the presence of 2 mM ATP and 10 mM MgCl$_2$. Where indicated, $E_1S$ or topotecan was added at 50 μM. The concentration of inorganic phosphate released by the hydrolysis of ATP was measured by tracking absorbance at 850 nm following the classical molybdate method[54]. ATPase rates were determined using linear regression in GraphPad Prism v8 and v9.

For transport assays, proteoliposomes in transport buffer (25 mM HEPES pH 7.5, 150 mM NaCl) were extruded through a 400 nm polycarbonate filter. MgCl$_2$ (5 mM) and $E_1S$ (50 μM, containing mixtures of $^3$H-$E_1S$ and $^1$H-$E_1S$) were added in the presence or absence of topotecan and the samples were incubated for 5 min at 30 °C. Transport reactions were initiated by adding ATP (2 mM) and stopped by adding an aliquot to ice-cold transport buffer containing unlabeled $E_1S$ (100 μM). Sample was filtered with a Multiscreen vacuum manifold (MSFBN6B filter plate, Millipore) and washed three times. Radioactivity trapped on the filters was measured with the microplate scintillation counter (Perkin Elmer 2450 Microbeta2). Data were analyzed and curves were plotted using the nonlinear regression Michaelis–Menten analysis and initial rates were calculated using the data points from the fitting curve (GraphPad Software, La Jolla, California, USA).

**Cryo-EM sample preparation**. All grids were prepared using a Vitrobot (FEI), with the environmental chamber set at 100% humidity and 4 °C. Nanodisc-reconstituted ABCG2 (1 mg ml$^{-1}$) was incubated with 5 mM ATP, 5 mM MgCl$_2$, 0.5 mM ADP, 100 μM topotecan or 200 μM $E_1S$ at room temperature for 10 min. 3.5 μl sample was applied on glow-discharged Quantifoil carbon grids (300 meshes, R 1.2/1.3 copper) immediately after incubation. Grids were blotted for 2.5 s with blot force 1 and flash-frozen in a mixture of liquid ethane and propane.

**Cryo-EM data acquisition**. Cryo-EM data of ABCG2-topotecan turnover sample was collected with a 300 keV Titan Krios (FEI) transmission electron microscope (TEM) equipped with a Gatan BioQuantum 1967 filter and a Gatan K3 camera. Images were recorded with three exposures per hole using EPU 2 in super-resolution mode with a 20 eV slit width of the energy filter and at a nominal magnification of 130,000x, resulting in a calibrated super-resolution pixel size of 0.33 Å. Defocus was set to vary from −0.6 to −2 μm. Each image was dose fractionated to 40 frames with 1.01 s total exposure time. The dose was 1.45 e$^-$/Å$^2$/frame (Total dose 58 e$^-$/Å$^2$). The super-resolution micrographs were down-sampled twice by Fourier cropping (to a pixel size of 0.66 Å), drift-corrected and dose-weighted using MotionCor2[55]. Micrographs were visually inspected, and bad micrographs were removed manually. The topotecan turnover dataset was composed of 24,177 movies from two sessions after removing bad micrographs.

Cryo-EM data of ABCG2-$E_1S$ turnover sample was collected on a 300 keV Titan Krios TEM equipped with a K2 Summit direct electron detector and a Quantum-LS energy filter (GIF) (20 eV zero-loss filtering; Gatan Inc.). The microscope was operated at a nominal magnification of 165,000× (the actual magnification 60,975×). Automated data collections were performed using SerialEM with seven exposures per hole using image-shift and coma-free setup[56]. Dose-fractionated movies were recorded in counting mode with a physical pixel size of 0.82 Å and the defocus was set in a range from −0.8 to −2.8 μm. Each movie was dose fractionated to 40 frames with 10 s total exposure time. The total dose was 50 e$^-$ /Å$^2$. The recorded movie stacks were initially processed by MotionCor2 (FOCUS integrated), including pre-processing procedures such as the gain-normalization, motion correction, and dose weighting[57,58]. The entire dataset of ABCG2-$E_1S$ turnover sample from two EM sessions was composed of 15,460 movies. Cryo-EM data collection statistics in this study are presented in Supplementary Table 3.

**Image processing**. The ABCG2-$E_1S$ turnover data were processed separately with a similar procedure until 2D classifications (Supplementary Fig. 2). The aligned micrographs were manually sorted in FOCUS[57] and imported to CryoSPARC 2.1[31]. The contrast transfer function (CTF) and defocus values were estimated on the dose-weighted micrographs using patch-based CTF in CryoSPARC. Images with poor quality were discarded. Defocus was calibrated with a range of −0.8 to –2.8 μm. Images with a CTF-estimated resolution of better than 5.2 Å were selected, resulting in a total of 11,343 micrographs. A 2D template was produced by initial 2D classifications with manually picked particles. Particles of the full dataset were auto-picked with 2D templates. After multiple rounds of 2D classification, particles were combined for 3D classification. The merged data with 645,803 particles were subjected to three rounds of 3D classification to classify distinct conformations. A 3D reference was generated from ab-initio 3D reconstructions. Each round was followed by a heterogenous refinement. As shown in Supplementary Fig. 2, several setting parameters were tested and applied (Round 1: 2 classes, 0.1 similarity, force hard-classification; Round 2: 2 classes, 0.8 similarity, force hard-classification; Round 2: 3 classes, 0.1 similarity, force hard-classification), resulting in a particle subset containing 221,160 particles. The best-resolved 3D class was further subjected to 3D nonuniform refinement (CryoSPARC). The map was refined with C1 symmetry by local refinement using a soft-mask. The overall resolution was 3.4 Å at FSC = 0.143 cutoff. A local-resolution map was generated by MonoRes[59].

For the topotecan turnover sample, the drift corrected, dose-weighted micrographs were imported in CryoSPARC v2.15[31]. Contrast transfer function

(CTF) parameters were estimated with GCTF[60]. The calculated defocus parameters of the micrographs were −0.3 to −2.7 μm. 1,534 particles were manually picked from selected micrographs to generate a 2D template for auto-picking. 2,623,169 particles were picked and extracted from 23,574 micrographs. After 10 rounds of classifications, 1,486,797 particles belonging to 'good' 2D classes were selected. An initial model was generated in Ab-Initio Reconstruction and used as a reference for the first round of classification. The selected particles from 2D classification were subjected to 3D classification with 4 classes and binned 3×. A good class from the first round of 3D classification with 750,851 particles was selected. This class was refined to 3.8 Å in Homogeneous refinement. The class was subjected to a second round of 3D classification with three classes and binned 3×. A good class with 455,604 particles was selected, and it was refined to 3.45 Å in Homogeneous refinement. A third round of 3D classification was done with 455,604 particles and three classes. An excellent class with 311,273 particles was selected, and it was refined to 3.34 Å in Homogeneous refinement. To check if resolution can be further improved by 3D classification, a fourth round of 3D classification was done with three classes and binned 3×. Two good classes with 299,139 particles were selected and refined to 3.28 Å in Homogeneous refinement. The 299,139 particles subset was subjected to Global CTF Refinement followed by Local CTF refinement. Further 3D variability analysis showed there was heterogeneity in one dimension for this particle subset. To deal with the heterogeneity, a fifth-round of 3D classification was performed with 3 classes and no bin. A minor class with 78,323 particles was isolated with the NBDs in a semi-closed conformation. Further homogeneous, local refinement and post-processing of the minor 3D class resulted in a 3.4 Å map named topotecan turnover-2, with automatically determined B factor of −92.8 Å$^2$. The other two major classes with 220,816 particles showed NBD open conformation, thus they were combined. Further homogeneous refinement, local refinement, and post-processing resulted in a 3.1 Å map named topotecan turnover-1, with automatically determined B factor of −106.5 Å$^2$. The final maps were refined in C1 symmetry and were directly used for modeling. Local resolution maps were generated using CryoSPARC v2.15. The data processing pipeline is shown in Supplementary Fig. 3.

A 3D variability analysis (3DVA) of the topotecan turnover sample was performed with particle stacks that contributed to high-resolution 3D maps in cryoSPARC v2.15[31]. The aim was to explore heterogeneity in single-particle cryo-EM data sets. A mask without lipids belt, three eigenvectors and 4 Å low-pass filtered were applied for 3DVA, generating simple linear "movies" of volumes. 3DVA outputs were visualized with 3D Variability Display tool in cryoSPARC[32].

We observed both turnover-1 and turnover-2 conformations in the topotecan turnover sample, whereas we only observed the turnover-2 conformation in the E$_1$S turnover sample. To avoid bias derived from reference map when doing 3D classification, we re-classified particles of the E$_1$S turnover sample into 6-10 classes using turnover-1 as a reference map. We expected to classify different conformations in this way. We observed some minor classes with separated NBDs. However, none of these classes produced EM maps of sufficient resolution to identify secondary structures. We can therefore exclude that the turnover-1 state with well-defined features is present in meaningful amounts in the E$_1$S turnover sample. Turnover-2 is the only well-defined state of the E$_1$S turnover sample.

**Model building and refinement.** Coot 0.9 was used for all model building steps[61]. ABCG2-MZ29 model (PDB 6FFC) was docked into turnover maps and used as the reference for manual rebuilding[20]. The coordinate of topotecan was from ABCG2-topotecan-Fabs structure (PDB 7NEZ). E$_1$S coordinate was from ABCG2-E$_1$S-Fab (PDB 6HCO). Cholesterol coordinate was from ABCG2-MZ29-Fab (PDB 6HIJ) and ATP coordinate was ABCG2$_{E211Q}$-ATP (PDB: 6HBU)[21]. The ligands were fitted into the EM density in Coot 0.9. We generated the restraints in eLBOW of Phenix[62]. Both topotecan and E$_1$S turnover structures were refined against their final maps respectively in real space refinement of Phenix[63]. In the final refinement, reciprocal-space refinement of the B factors and minimization global refinements were applied together with standard geometry, rotamer, Ramachandran plot, Cβ, non-crystallographic symmetry and secondary structure restraints. The quality of final model was assessed by MolProbity[64]. The refinement statistics are in Supplementary Table 3.

For model validation, we applied 0.3 Å random shifts to final models using phenix_pdb_tools[65]. The scrambled model was refined against one of the unfiltered half maps (half map A). The Fourier shell correlation between refined scrambled model and half map A was plotted as FSC$_{work}$. The Fourier shell correlation between refined scrambled model and half map B was plotted as FSC$_{free}$. The overlay between the FSC$_{work}$ and FSC$_{free}$ indicated no over-fitting presence.

**Figure preparation.** Figures were prepared with PyMOL (The PyMOL Molecular Graphics System, Version 2.4.0 Schrödinger, LLC), GraphPad Prism v8, v9, and UCSF ChimeraX[66]. Movies were prepared with UCSF Chimera[67]. The volume of cavity 1 in ABCG2 turnover structures was determined using POCASA[44].

## Data availability

Atomic coordinates for the E$_1$S-bound turnover-2 structure and the topotecan-bound turnover-1 and turnover-2 structures are deposited with the Protein Data Bank under accession codes 7OJ8, 7OJH, and 7OJI, respectively. The cryo-EM density maps for the three structures were deposited in the Electron Microscopy Data Bank under accession codes EMD-12939 (E1S-bound turnover-2 state), EMD-12951 (topotecan-bound turnover-1 state) and EMD-12952 (topotecan-bound turnover-2 state). Source Data for Figs. 1a, 3e and 3f, Supplementary Figures 1a, 2e, 3f, 3g, 8a, 9a, and 9b are provided with this paper in the Source Data file. Raw EM data are available upon request. A Life Sciences Reporting Summary for this article is available. Source data are provided with this paper.

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

## Acknowledgements

This research was supported by the Swiss National Science Foundation through the National Centre of Competence in Research (NCCR) TransCure. We thank the Scientific Center for Optical and Electron Microscopy (ScopeM) at ETH Zürich for technical support. We thank K. Goldie, R. Irobalieva, L. Kováčik and A. Fecteau-Lefebvre for technical support with EM data collection.

## Author contributions

Q.Y. and I.M. expressed and purified wild-type ABCG2. Q.Y. cloned, expressed, and purified the ABCG2$_{R184A}$ variant. S.M.J. and I.M. reconstituted ABCG2 in lipid nanodiscs and prepared EM grids of the E$_1$S turnover sample with the help of J.K. Q.Y. reconstituted ABCG2 in lipid nanodiscs and prepared EM grids of topotecan turnover sample with the help of J.K. Q.Y. reconstituted ABCG2 into liposomes and carried out all functional experiments. D.N. collected and processed cryo-EM data and determined the structure of the E$_1$S turnover sample supervised by H.S. J.K. collected cryo-EM data of the topotecan turnover sample. Q.Y. processed EM data of the topotecan turnover ABCG2 sample and determined the turnover-1 and turnover-2 structures with the help of J.K. D.N. built and refined initial model of the E$_1$S turnover structure. K.P.L. and Q.Y. built, refined, and validated the structures. K.P.L. and Q.Y. wrote the manuscript with input from all authors. All authors contributed to revision of the manuscript.

## Competing interests

The authors declare no competing interests.

## Additional information

Peer review information Nature Communications thanks the anonymous reviewers for their contributions ot the peer review of this work. Peer review reports are available.

