## [Peer Review File · Nature Communications]

REVIEWER COMMENTS

Reviewer #1 (Remarks to the Author):

In the manuscript “Structures of ABCG2 under turnover conditions reveal a key step in drug transport mechanism” the authors provide high-resolution structures of two different states of the important ABC transporter ABCG2.

Following a general trend in the ABC transporter field, the authors intentionally activate the transporter prior to vitrification (through the addition of ATP and substrate – in this study two distinct substrates were used), instead of locking it in a respective state, as previously done for all ABCG2 structures. This strategy led to previously undescribed states which complement the transport cycle of the membrane protein.

The structures are of sufficient quality for publication and the manuscript is of broad general interest.

I only have one major concern that needs to be relieved before publication can be considered:

For all three structures, the authors report the presence of two ATPs at their respective binding sites. No alternative conformations/ nucleotide states were found in the dataset. This finding is very interesting, as it is entirely unexpected. It indicates, that the double ATP bound state is (by far) the state with the lowest energy. This contradicts the fact that ATP binding is necessary to convert the transporter from an inward open to an outward open state - providing the first power stroke. Therefore, it should be a state of high energy.

In a turnover experiment, I would anticipate multiple states with various nucleotide states. If the authors claim that they have analyzed a dynamic system – I expect to see some dynamics in the preparation. From my perspective, it is far more likely that the transporter is constrained by the nanodisc or the substrate, or possibly that the protein preparation was once somehow faulty (or of course a combination). Following the authors' argument (on page 5: the authors expected that the substrate would be too mobile to be visualized at high resolution, or not present at all) something might have happened to the preparation that constraints substrate transport.

What would happen if the transporter was analyzed in detergent, or without substrate in nanodisc? Did the authors perform an ATPase activity assay with exactly the same preparation that was used for the cryo-EM preparation?

The finding of an exclusive double ATP-bound state is the central point of the manuscript. However, this is highly unexpected and unusual. Therefore, additional controls (as mentioned above) and extra care must be taken.

Minor comments:

An article is missing in the headline – a key step in the drug transport mechanism

Reviewer #2 (Remarks to the Author):

This manuscript details the structural and functional characterization of ABCG2 under turnover conditions in the presence of transported substrate and physiological concentrations of MgATP and MgADP. It represents a significant advance in the mechanistic understanding of this family of ABC transporters, based on the following observations:

1. identification of two distinct conformational states where both the transport substrate and MgATP are bound. These states, termed turnover-1 and turnover-2, are distinct inward facing conformations; turnover-1 has widely separated NBDs and an accessible substrate binding cavity, while turnover-2 has semi-closed NBDs and an almost fully occluded cavity between the TMDs.
2. The transition from turnover-1 to turnover-2 is associated with changes in the "VSGGE" loop (the variant of "LSGGQ" present in ABCG2) that help stabilize TMD closure in the turnover-2 state - thus linking ATP binding and TMD closure.
3. Inhibitors of ABCG2 are characterized by steric clashes that block the transition from turnover-1 to turnover-2 states.

Based on this analysis, it is proposed that the transition from turnover-1 to turnover-2 is the rate-limiting step of the reaction cycle, where true substrates can navigate this conformational landscape and complete the transport cycle, while inhibitors cannot. From a quantitative analysis of the ATPase and transport rates, it is concluded that ABCG2 may require multiple attempts (each consuming ATP) before substrate extrusion is successfully accomplished - an important observation.

Despite the extensive structural analyses of ABC transporters in various conformational states, structures with both substrate and ATP bound have been conspicuously absent. To my knowledge, the only other example where both the binding of both substrate and ATP can be modeled is the regulatory ABC transporter sulfonyleurea receptor 1 of ATP-sensitive potassium (KATP) channels (Martin et al. eLife 2019;8:e46417), which is a channel and not a true transporter. Hence, the present work is a major accomplishment not only for the ABCG2 system, but also for understanding how substrate binding and translocation are linked to ATP binding and hydrolysis.

A few comments for the authors considerations

page 9 - last paragraph - the ATPase rates appear to have an editing error in the units (molecules phosphate per ABCG2 dimer "per second", rather than "and second")

supplementary table 1 - the number of significant figures in the RMSD seems excess given the coordinate accuracy at $\sim 3 \text{ \AA}$ resolutions.

Figure 3 c/d - is Mg²⁺ present in the binding site? (it doesn't look like it in the figures)

Reviewer #3 (Remarks to the Author):

Yu Q. et al perform nice studies with cryo-EM to determine structures of either estrone-1-sulfate-bound ABCG2 or Topotecan-bound ABCG2 under turnover conditions (ie, ATP, ADP MgCl₂ without the EL3 binding monoclonal antibody 5D3). These substrates are representative of an rarely used endogenous substrate, estrone-1-sulfate (E-1-S, ~350 dalton) and the exogenous substrate topotecan (~458 dalton). Unexpectedly, the authors found substrates were bound along with two ATP bound molecules (although it remains unclear and needs to be clarified how they know ATP is bound vs ADP or a combination of ATP and ADP). Importantly they observe two distinct conformational states, one state turnover-1 and the other is turnover state-2. The larger substrate can be observed in turnover state-1 and -2 whereas the smaller molecular weight substrate, E-1-S is only observed in turnover-2, however, it is unclear how one can assert turnover state-1 is not observed when their density maps used cannot resolve this . Perhaps some caveats should be applied to the interpretation.

The authors seem to assert that the binding cavity did not collapse even when ATP is bound. Why is this so different than the previously reported structure of E1S-5D3-Fab-bound-ABCG2. The authors might clarify if substrates change their contacts (especially topotecan) with residues in the binding during the transport cycle shown in these structures.

The relationship between the structure and the catalytic activity needs to be reconciled. Specifically, in the turnover-1 state of the topotecan bound structure (found in 88% of the particles), has the NBDs widely separated which seems consistent with a basal ATPase rate that is decreased by topotecan. However, for the proteoliposome-reconstituted ABCG2, has increased basal ATPase in the presence of topotecan (Fig. 3e). Furthermore, this is different than in the nanodisc ATPase of reconstituted ABCG2 which appears suppressed by ABCG2. Please account for these biochemical differences in the context of the structures.

In the results (page 5), the authors write “We also did not observe a collapsed apo-state similar to that reported recently for apo-ABCG2 (Ref. 22)”. What does this statement mean exactly? More detailed information is needed.

In the result (page 5), they described “In all turnover structures, the leucine gate (formerly called leucine plug), which separates cavity 1 from cavity 2, is closed (Supplementary Figs. 4a, 4b)”. Could you provide close-up view to show that the leucine gate (L554 and L555; distance) is actually closed? Does it not change at all going from turnover-1 to turnover-2?

Comparison of the topotecan-bound turnover 1 and turnover 2 revealed residues from Q181 to V186 help facilitate the transition from turnover-1 to turnover-2 through a conformational change.. Mutation of a single residue, R184, produced decreased ATPase and transport activities. It seems that residues 181-186 stabilize the TDM domain closure in turnover 2, but not in turnover 1. Importantly, R184A substitution showed reduced E1S transport and ATP hydrolysis with either E1S or topotecan versus WT. The authors assert that “the mutation of R184 to alanine neither interfered with substrate binding nor abolished the NBD-TMD coupling”. However, the data seems insufficient to reach draw this conclusion. To claim this, the authors need to determine the EC₅₀ of substrate-induced stimulation of the ATPase activity in R184A and/or assess transport binding affinity (K_m). Further, the reduction of the initial E1S transport rate is not enough to conclude that “topotecan and E1S bind and compete for the same binding pocket both in WT ABCG2 and in the ABCG2R184A variant” (page 7). To show competitive inhibition the authors would need to perform kinetic analysis of transport inhibition by linear and nonlinear regression (e.g. Lineweaver-Burk and Eadie-Hofstee) in R184A.

In the discussion (page 9), they described “If ABCG2 hydrolyses two molecules of ATP in one productive transport cycle, the net futile ATPase rate in the presence of E1S is 1.5 molecules phosphate per ABCG2 dimer and second, which is 2.5 times higher than in the absence of E1S”. The authors need to clarify the details of their calculations and how they arrived at this.

The authors determine steric clashes between the binding cavities in turnover 1 and 2 and the figure 5 caption states “inhibitor structures were extracted from inward-open ABCG2 structures containing these inhibitors (PDB 6-ETI and 6-FEQ, respectively) and manually placed into cavity 1 of the topotecan turnover-2 structure. I think “manually placed.” Is imprecise and there is no reference to docking studies or even RMSD alignment. Including this information into the methods rather than leaving it as a quick aside in a caption might strengthen their steric argument.

In the Discussion the author assert, based on a smaller volume of cavity 1 during the transition from turnover 1 to turnover 2, that “small molecule inhibitors ... block the transition from turnover 1 to turnover 2”. In the absence of any data to support this (no small molecule inhibitor structures have been reported to my knowledge), this seems to be a bold assertion that either needs to be supported by data or substantially modified.

Writing and figure quality

Figures are good aesthetically with consistent application of color. The manuscript is relatively easy to read and follow. The introduction section could be bolstered by swapping out the phrase “this was,” used 3 times across page 3, for a more specific phrase to alert the reader which out of the previous information “this” refers to. The methods section also contains some obvious grammatical errors that require attention.

Reviewer responses NCOMMS-21-08286

We thank the reviewers for their encouraging and insightful comments and have made numerous changes in the manuscript to address the concerns raised. In the following, reviewer comments are in italics while our actions in response are described in plain text.

Reviewer #1:

In the manuscript "Structures of ABCG2 under turnover conditions reveal a key step in drug transport mechanism" the authors provide high-resolution structures of two different states of the important ABC transporter ABCG2. Following a general trend in the ABC transporter field, the authors intentionally activate the transporter prior to vitrification (through the addition of ATP and substrate – in this study two distinct substrates were used), instead of locking it in a respective state, as previously done for all ABCG2 structures. This strategy led to previously undescribed states which complement the transport cycle of the membrane protein. The structures are of sufficient quality for publication and the manuscript is of broad general interest. I only have one major concern that needs to be relieved before publication can be considered:

We split the reviewer's concern into four points:

1. For all three structures, the authors report the presence of two ATPs at their respective binding sites. No alternative conformations/ nucleotide states were found in the dataset. This finding is very interesting, as it is entirely unexpected. It indicates, that the double ATP bound state is (by far) the state with the lowest energy. This contradicts the fact that ATP binding is necessary to convert the transporter from an inward open to an outward open state - providing the first power stroke. Therefore, it should be a state of high energy.

We were just as surprised as the reviewer when we processed our data. However, there isn't necessarily a contradiction. The transporter is turning over until the solution is frozen during grid preparation. In our proposed mechanistic scheme, one could argue that all other states are of higher energy in the presence of transport substrates and ATP. There is no contradiction with any of the earlier / published work.

2 In a turnover experiment, I would anticipate multiple states with various nucleotide states. If the authors claim that they have analyzed a dynamic system – I expect to see some dynamics in the preparation.

In our paper we did not claim to study the dynamics of the system, nor did we state that we are studying "a dynamic system." Instead, we study ABCG2 under turnover conditions. There is a fine but important difference between these two concepts. To analyze the dynamics of ABCG2, alternative methods (e.g. spectroscopy such as DEER) would be required, which is outside of the premise of our study. In our study, we searched for classes of particles that allowed high-resolution structure determination. While other possible conformations/states are undoubtedly present, they are (i) much less populated and (ii) less well-ordered under turnover condition, so we did not obtain high-resolution structure for such possible conformations/states.

3. From my perspective, it is far more likely that the transporter is constrained by the nanodisc or the substrate, or possibly that the protein preparation was once somehow faulty (or of course a combination). Following the authors' argument (on page 5: the authors expected that the substrate would be too mobile to be visualized at high resolution, or not present at all) something might have happened to the preparation that constraints substrate transport.

Our lab has done intensive work on ABCG2, and our protein preparations are routinely tested for function (transport across liposomes membranes and ATPase modulation in liposomes and nanodiscs). We are therefore very confident that our protein preparations are not "faulty". Nanodiscs are widely regarded as the current gold standard for structural studies of membrane proteins. They allow for high-resolution structure

determination in an environment that contains lipid mixtures that most closely resemble the physiological environment.

4. What would happen if the transporter was analyzed in detergent, or without substrate in nanodisc?

These are important considerations that we address as follows:

- Detergents: ABCG2 shows extremely low ATPase activity in DDM:CHS, which is by far the best detergent mixture for ABCG2 solubilization and purification. Importantly, we have investigated the modulation of ABCG2 ATPase activity by substrates and inhibitors and found that it is almost completely absent in detergent (see figure to the right). This suggests that detergent molecules interfere with substrate binding, making it a very poor environment for structural studies. For this reason, nanodiscs are the preferred environment.

- "Apo" conditions: There are two main reasons that strongly argue against such an experiment: First: In order to investigate the transporter under turnover conditions, all relevant components need to be present. It defeats the purpose of the study to leave out one of the critical components (the transport substrate). Second, in the absence of transport substrates, ABCG2 in nanodiscs shows very high basal ATPase activity when compared with ABCG2 in liposomes. We have reported this as early as 2017 (Taylor *et al.*, *Nature* **546**, 504-509, 2017). It was concluded at the time that cholesterol molecules were present in the substrate binding pocket (Suppl. Fig. 8 of the Taylor 2017 paper). For this reason, studying ABCG2 in the absence of transport substrate may result in artifacts. In contrast, we found that when substrates are added at sufficient concentration, the observed ATPase rates of ABCG2 in nanodiscs and liposomes are more similar (new biochemical data shown in the figure below).

Given that there is always a certain error associated with protein concentration determination, the differences at saturating drug concentration are even less significant. Importantly, the data reveal similar EC₅₀ values both for E₁S and topotecan. This speaks strongly in favor of conducting our experiments at saturating drug concentrations, which is precisely what we have done.

In conclusion, whereas our study captures ABCG2 in functionally relevant conditions, studying ABCG2 in nanodiscs in the absence of transport substrate is unlikely to contribute anything more meaningful to understanding the reaction cycle.

5. Did the authors perform an ATPase activity assay with exactly the same preparation that was used for the cryo-EM preparation?

Yes, we performed ATPase activity assay with representative substrates and inhibitors and checked the sample on SDS-page for every preparation we used for cryo-EM analysis.

6. The finding of an exclusive double ATP-bound state is the central point of the manuscript. However, this is highly unexpected and unusual. Therefore, additional controls (as mentioned above) and extra care must be taken.

While we agree that the findings are remarkable, we were extra careful with all steps of the functional and structural studies.

Minor comments:

An article is missing in the headline – a key step in the drug transport mechanism

Corrected.

Reviewer #2:

This manuscript details the structural and functional characterization of ABCG2 under turnover conditions in the presence of transported substrate and physiological concentrations of MgATP and MgADP. It represents a significant advance in the mechanistic understanding of this family of ABC transporters, based on the following observations:

1. identification of two distinct conformational states where both the transport substrate and MgATP are bound. These states, termed turnover-1 and turnover-2, are distinct inward facing conformations; turnover-1 has widely separated NBDs and an accessible substrate binding cavity, while turnover-2 has semi-closed NBDs and an almost fully occluded cavity between the TMDs.

2. The transition from turnover-1 to turnover-2 is associated with changes in the "VSGGE" loop (the variant of "LSGGQ" present in ABCG2) that help stabilize TMD closure in the turnover-2 state - thus linking ATP binding and TMD closure.

3. Inhibitors of ABCG2 are characterized by steric clashes that block the transition from turnover-1 to turnover-2 states.

Based on this analysis, it is proposed that the transition from turnover-1 to turnover-2 is the rate-limiting step of the reaction cycle, where true substrates can navigate this conformational landscape and complete the transport cycle, while inhibitors cannot. From a quantitative analysis of the ATPase and transport rates, it is concluded that ABCG2 may require multiple attempts (each consuming ATP) before substrate extrusion is successfully accomplished - an important observation. Despite the extensive structural analyses of ABC transporters in various conformational states, structures with both substrate and ATP bound have been conspicuously absent. To my knowledge, the only other example where both the binding of both substrate and ATP can be modeled is the regulatory ABC transporter sulfonylurea receptor 1 of ATP-sensitive potassium (KATP) channels (Martin et al. eLife 2019;8:e46417), which is a channel and not a true transporter. Hence, the present work is a major accomplishment not only for the ABCG2 system, but also for understanding how substrate binding and translocation are linked to ATP binding and hydrolysis.

We thank the reviewer for this supportive comment.

A few comments for the authors considerations:

1. page 9 - last paragraph - the ATPase rates appear to have an editing error in the units (molecules phosphate per ABCG2 dimer "per second", rather than "and second")

Corrected.

2. supplementary table 1 - the number of significant figures in the RMSD seems excess given the coordinate accuracy at ~ 3 Å resolutions.

We agree and have reduced the number of significant digits.

3. Figure 3 c/d - is Mg²⁺ present in the binding site? (it doesn't look like it in the figures)

Based on the EM maps, we cannot identify significant density for Mg²⁺ at the current resolution, as shown in the figure below. For this reason, we did not build Mg²⁺ in the models.

Reviewer #3:

Yu Q. et al perform nice studies with cryo-EM to determine structures of either estrone-1-sulfate-bound ABCG2 or Topotecan-bound ABCG2 under turnover conditions (ie, ATP, ADP MgCl₂ without the EL3 binding monoclonal antibody 5D3). These substrates are representative of an rarely used endogenous substrate, estrone-1-sulfate (E-1-S, ~350 dalton) and the exogenous substrate topotecan (~458 dalton).

1. Unexpectedly, the authors found substrates were bound along with two ATP bound molecules (although it remains unclear and needs to be clarified how they know ATP is bound vs ADP or a combination of ATP and ADP).

We thank the reviewer for this important question. Given that we indeed use a mixture of ATP and ADP to mimic physiological conditions, we were faced with a decision on how to interpret the observed density. In the figure below we show the EM density of the topotecan-bound turnover-2 state at each nucleotide binding site using a contour level suitable for side chain model building. We then docked either ATP or ADP into the density corresponding to the nucleotide.

The analysis suggests that two ATP molecules fit the density best, while unexplained additional density would be present if ADP were modeled. When increasing the sigma level further, the density for the three phosphate groups disappears simultaneously, arguing against a mixture of ADP and ATP.

In the EM map of turnover-1, the occupancy of two nucleotides is generally weaker because there are fewer contacts to the protein. The figure to the right shows the relevant section of the map at three distinct levels. Based on the presence of substrates and the separated NBD conformation, we speculate that the transporter is in pre-translocation state. While it would in principle be possible to build a mixture of ADP and ATP in this case, we built ATP because it didn't violate the observed density features and because the concentration of ATP in the solution is 10x higher (and its affinity likely higher than that of ADP, in analogy to findings with other ABC transporters). We have added these considerations to the manuscript.

2. Importantly they observe two distinct conformational states, one state turnover-1 and the other is turnover state-2. The larger substrate can be observed in turnover state-1 and -2 whereas the smaller molecular weight substrate, E-1-S is only observed in turnover-2, however, it is unclear how one can assert turnover state-1 is not observed when their density maps used cannot resolve this. Perhaps some caveats should be applied to the interpretation.

This is a very important point. We had discussed this issue in the Methods section of the manuscript (page 13 lines 479-487). We indeed attempted to re-classify particles for both datasets using turnover-1 as reference map and analyzed the conformations of resulting classes. Below we

show a histogram of the conformation distribution for each dataset. The predominant class in the E₁S turnover sample is turnover-2, whereas the predominant class in the topotecan turnover sample is turnover-1.

As discussed in the paper, we were able to determine a high-resolution structure of the turnover-2 state in the topotecan sample. In contrast, while we did observe 3D classes featuring separated NBDs in the E₁S-containing sample, neither of these classes led to high-resolution structures. We can therefore exclude that a turnover-1 state with well-defined features is present in meaningful amounts in the E₁S turnover sample. The figure below shows a representative map of the NBD-open class in the E₁S turnover sample.

3 The authors seem to assert that the binding cavity did not collapse even when ATP is bound. Why is this so different than the previously reported structure of E₁S-5D3-Fab-bound-ABCG₂.

This question refers to a previously published paper. In the structure of E₁S-5D3-Fab-bound-ABCG₂^{E211Q} (Manolaridis *et al.*, *Nature* **563**, 426-430, 2018), the inhibitory 5D3-Fab was present. Density for ATP was not observed in the nucleotide-binding pockets even though ATP had been added to the sample. We conclude that the presence of 5D3-Fab prevented the sample from reaching a closed conformation or a turnover state. When no 5D3-Fab was present, the ABCG₂^{E211Q} variant adopted a closed conformation with a collapsed translocation pathway (same 2018 paper). The key difference to our present study was that the previous samples were not under turnover conditions. We conclude that while substrate is bound to ABCG₂, the translocation pathway cannot assume a collapsed conformation due to steric clashes.

4. The authors might clarify if substrates change their contacts (especially topotecan) with residues in the binding during the transport cycle shown in these structures.

Thank you for pointing out this issue. Based on the EM maps of turnover-1 and topotecan- and 5D3- Fab-bound inward-open conformation, the substrate did not change their contacts with residues in the binding cavity. Given the relatively large density feature, several ways of fitting topotecan are possible. In the original version of the manuscript, readers might indeed have got the impression that topotecan greatly changed its contacts. This was due to a slightly misleading way we had used in the initial version of the figure. In our revision, we have updated Fig. 1 with respect to the fitting of topotecan in turnover-1 so that it is fitted the same way as in previously structures and thus consistency is maintained.

5. The relationship between the structure and the catalytic activity needs to be reconciled. Specifically, in the turnover-1 state of the topotecan bound structure (found in 88% of the particles), has the NBDs widely separated which seems consistent with a basal ATPase rate that is decreased by topotecan. However, for the proteoliposome-reconstituted ABCG2, has increased basal ATPase in the presence of topotecan (Fig. 3e). Furthermore, this is different than in the nanodisc ATPase of reconstituted ABCG2 which appears suppressed by ABCG2 [We assume the reviewer means topotecan here]. Please account for these biochemical differences in the context of the structures.

We agree and have both added new biochemical data and modified our discussion to better explain this point. In the figure below, we titrated the two substrates E1S and topotecan and measured the ATPase rate of nanodisc- vs. liposome-reconstituted ABCG2.

In the absence of transport substrates, ABCG2 in nanodiscs shows very high basal ATPase activity when compared with ABCG2 in liposomes. We have reported this as early as 2017 (Taylor et al., *Nature* **546**, 504-509, 2017). It was concluded at the time that cholesterol molecules were present in the substrate binding pocket (Suppl. Fig. 8 of the Taylor 2017 paper). For this reason, determining a structure of ABCG2 in the absence of transport substrate may result in artifacts, and incorporating the basal ATPase of apo-ABCG2 levels into a mechanistic schema is more meaningful using liposome-based data rather than nanodisc-based data.

We found that when substrates are added at sufficient concentration, the observed ATPase rates of ABCG2 in nanodiscs and liposomes are more similar. Importantly, the data reveal similar EC₅₀ values both for E₁S and topotecan. This speaks strongly in favor of conducting our experiments at saturating drug concentrations, which is precisely what we have done.

6. In the results (page 5), the authors write “We also did not observe a collapsed apo-state similar to that reported recently for apo-ABCG2 (Ref. 22)”. What does this statement mean exactly? More detailed information is needed.

Our statement refers to a recently published paper by Orlando et al³ (Ref. 22), where the authors reported an apo-ABCG2 structure with collapsed TMD domains and separated NBD domains. We did not observe this conformation in our turnover study. As argued above, determining a structure of nanodisc-reconstituted ABCG2 in the absence of substrates but the presence of ATP/ADP/Mg²⁺ may result in artifacts because the ATPase rates are artificially high. In our paper we have to reference the study by Orlando et al, even though we believe the conformation observed by them is an artifact.

7. In the result (page 5), they described “In all turnover structures, the leucine gate (formerly called leucine plug), which separates cavity 1 from cavity 2, is closed (Supplementary Figs. 4a, 4b)”. Could you provide close-up view to show that the leucine gate (L554 and L555; distance) is actually closed? Does it not change at all going from turnover-1 to turnover-2?

Here we added the closeup view of leucine gate with distance labeled. Both leucine gates are closed in turnover-1 and turnover-2. There are very minor structural changes that are not significant given the resolution of our 3D reconstructions.

8. Comparison of the topotecan-bound turnover 1 and turnover 2 revealed residues from Q181 to V186 help facilitate the transition from turnover-1 to turnover-2 through a conformational change. Mutation of a single residue, R184, produced decreased ATPase and transport activities. It seems that residues 181-186 stabilize

the TDM domain closure in turnover 2, but not in turnover 1. Importantly, R184A substitution showed reduced E1S transport and ATP hydrolysis with either E1S or topotecan versus WT. The authors assert that “the mutation of R184 to alanine neither interfered with substrate binding nor abolished the NBD-TMD coupling”. However, the data seems insufficient to reach draw this conclusion. To claim this, the authors need to determine the EC₅₀ of substrate-induced stimulation of the ATPase activity in R184A and/or assess transport binding affinity (K_m).

We agree and include additional biochemical data (Suppl. Fig. 9a, see Figure below) where we determined the EC₅₀ of substrate-induced stimulation of the ATPase activity of WT ABCG2 and the variant R184A. The EC₅₀ values of topotecan-induced ATPase stimulation in liposome-reconstituted is ~12μM for WT ABCG2 and ~6μM for ABCG2_{R184A}. The difference is very small, in particular when considering the experimental errors and the fact that V_{max} in ABCG2_{R184A} is much lower. The situation is similar with E₁S: The EC₅₀ values of E₁S-induced ATPase stimulation is ~12μM both for WT ABCG2 and ABCG2_{R184A}. We believe this supports our conclusions.

9. Further, the reduction of the initial E₁S transport rate is not enough to conclude that “topotecan and E₁S bind and compete for the same binding pocket both in WT ABCG2 and in the ABCG2_{R184A} variant” (page 7). To show competitive inhibition the authors would need to perform kinetic analysis of transport inhibition by linear and nonlinear regression (e.g. Lineweaver-Burk and Eadie-Hofstee) in R184A.

Here we disagree, as we did not call any of the compounds “competitive inhibitors.” We refer to structural findings. The structural data in this manuscript and in previously published papers (Manolaridis *et al.*, *Nature* **563**: 426-430, 2018; Orlando BJ & Liao M, *Nat Commun* **11**: 2264, 2020; Kowal J *et al.* *J Mol Biol* **433**: 166980, *epub April 7, 2021*,) demonstrate clearly that topotecan and E₁S bind to the same cavity in ABCG2.

10. In the discussion (page 9), they described “If ABCG2 hydrolyses two molecules of ATP in one productive transport cycle, the net futile ATPase rate in the presence of E₁S is 1.5 molecules phosphate per ABCG2 dimer and second, which is 2.5 times higher than in the absence of E₁S”. The authors need to clarify the details of their calculations and how they arrived at this.

We thank the reviewer for spotting our mistake. The calculations are as follows:

- ATPase rate of WT ABCG2 in the absence of E₁S: 270 nmol min⁻¹mg⁻¹, equivalent to 0.65 molecules ATP per ABCG2 dimer per second
- ATPase rate of WT ABCG2 stimulated by E₁S: 1.0 μmol min⁻¹mg⁻¹, equivalent to 2.4 molecules ATP per ABCG2 dimer per second
- E₁S transport rate at saturating E₁S concentration: 40nmol min⁻¹mg⁻¹, equivalent to 0.1 molecules E₁S per ABCG2 dimer per second (correction for orientation in liposomes applied)

The futile ATPase rate in the presence of saturating E₁S is therefore 1000-80=920 nmol min⁻¹mg⁻¹ (2x40 because 2 molecules ATP used for productive transport cycle). Expressed in molecules ATP hydrolyzed per ABCG2 dimer per second, the number is 2.4 - 0.2 = 2.2 Thus the futile ATPase rates in the presence of E₁S is 3.4 times higher than in its absence.

11. The authors determine steric clashes between the binding cavities in turnover 1 and 2 and the figure 5 caption states “inhibitor structures were extracted from inward-open ABCG2 structures containing these inhibitors (PDB 6-ETI and 6-FEQ, respectively) and manually placed into cavity 1 of the topotecan turnover-2 structure. I think “manually placed.” Is imprecise and there is no reference to docking studies or even RMSD alignment. Including this information into the methods rather than leaving it as a quick aside in a caption might strengthen their steric argument.

It appears we didn't make it clear to the reviewer that “manual placing” is not arbitrary but based on high-resolution structures of ABCG2 bound to these inhibitors (Jackson S *et al.*, *Nat Struct Mol Biol* **25**: 333-340, 2018). The pair of F439 sidechains (one from each ABCG2 monomer) are key residues to interact with inhibitors or substrates in the drug-binding cavity of ABCG2 (see also Manolaridis *et al.*, *Nature* **563**: 426-430, 2018; Orlando BJ & Liao M, *Nat Commun* **11**: 2264, 2020; Kowal J *et al.* *J Mol Biol epub April 7, 2021*, doi: 10.1016/j.jmb.2021.166980). To place inhibitors into cavity 1 of our turnover structures, we aligned the two F439s in PDB 6ETI or in 6FEQ with the F439 residues of our structures. In this way the translocation pathway of PDB 6ETI, 6FEQ and turnover-2 are well-aligned and inhibitors can be fit into cavity 1 of turnover-2. To make this clearer we updated the legend of Fig 5.

12. In the Discussion the author assert, based on a smaller volume of cavity 1 during the transition from turnover 1 to turnover 2, that “small molecule inhibitors ... block the transition from turnover 1 to turnover 2”. In the absence of any data to support this (no small molecule inhibitor structures have been reported to my knowledge), this seems to be a bold assertion that either needs to be supported by data or substantially modified.

The reviewer seems to have missed that there are in fact two published structures of ABCG2 bound to small-molecule inhibitors (Jackson S *et al.*, *Nat Struct Mol Biol* **25**: 333-340, 2018). The relevant paper is listed as reference 20 in our manuscript.

Writing and figure quality

Figures are good aesthetically with consistent application of color. The manuscript is relatively easy to read and follow. The introduction section could be bolstered by swapping out the phrase “this was,” used 3 times across page 3, for a more specific phrase to alert the reader which out of the previous information “this” refers to. The methods section also contains some obvious grammatical errors that require attention.

We have made an effort to improve the readability of the manuscript, but could not fully avoid certain repetitions.

REVIEWERS' COMMENTS

Reviewer #1 (Remarks to the Author):

The authors have addressed my concerns in great detail and to my satisfaction.

While I do have a different opinion on the different aspects of controls (e.g. detergent vs. nano disc - apo vs. substrate bound) - I do not think that this should preclude publication.

I would want to clarify that I never implied that the protein purifications of the lab are generally faulty - I said that a single prep was once faulty - which can obviously happen in every lab.

Reviewer #2 (Remarks to the Author):

The authors have satisfactorily addressed my concerns

Reviewer #3 (Remarks to the Author):

For the most part, I thank the authors for being very responsive to my queries. The one exception is point 12 regarding "small molecule inhibitors...block the transition from turnover1 to turnover 2". I am aware of the authors studies. My quibble with the statement is that one is inferring a small molecule is capable of blocking the transition from turnover one to turnover 2. This might be true, but in actuality it is an inference with no empirical data to support the notion that such a block occurs, consequently, this statement needs to be amended

Reviewer responses: We thank the reviewers for their comments and have made further changes in the manuscript to address the concerns raised. In the following, reviewer comments are in italics while our actions in response are in plain text.

Reviewer #1 (Remarks to the Author):

The authors have addressed my concerns in great detail and to my satisfaction.

While I do have a different opinion on the different aspects of controls (e.g. detergent vs. nano disc - apo vs. substrate bound) - I do not think that this should preclude publication. I would want to clarify that I never implied that the protein purifications of the lab are generally faulty - I said that a single prep was once faulty - which can obviously happen in every lab.

We thank the reviewer for the positive comment and the clarification regarding the question to sample preparation.

Reviewer #2 (Remarks to the Author):

The authors have satisfactorily addressed my concerns

We thank the reviewer for the positive comment.

Reviewer #3 (Remarks to the Author):

For the most part, I thank the authors for being very responsive to my queries. The one exception is point 12 regarding "small molecule inhibitors...block the transition from turnover1 to turnover 2". I am aware of the authors studies. My quibble with the statement is that one is inferring a small molecule is capable of blocking the transition from turnover one to turnover 2. This might be true, but in actuality it is an inference with no empirical data to support the notion that such a block occurs, consequently, this statement needs to be amended.

We agree with the reviewer and have made several changes to the relevant paragraph on page 8, lines 246-262 to avoid overinterpretation. We no longer claim any blocking / block occurs. Instead, we describe that binding of the inhibitors as they were found in the high-resolution structures is incompatible with the reduced size of the substrate-binding cavity (cavity 1) in the turnover-2 state.